# Microbiome Diversity and Dynamics in Lotus–Fish Co-Culture Versus Intensive Pond Systems: Implications for Sustainable Aquaculture

**DOI:** 10.3390/biology14081092

**Published:** 2025-08-20

**Authors:** Qianqian Zeng, Ziyi Wang, Zhongyuan Shen, Wuhui Li, Kaikun Luo, Qinbo Qin, Shengnan Li, Qianhong Gu

**Affiliations:** 1Engineering Research Center of Polyploid Fish Reproduction and Breeding of the State Education Ministry, College of Life Sciences, Hunan Normal University, Changsha 410081, China; zqq@hunnu.edu.cn (Q.Z.); 202130233024@hunnu.edu.cn (Z.W.); zyshen@hunnu.edu.cn (Z.S.); liwuhui11@163.com (W.L.); lkk202507@163.com (K.L.); qqb@hunnu.edu.cn (Q.Q.); 2Hunan Yuelu Mountain Science and Technology Co., Ltd., Aimed at Aquatic Breeding, Changsha 410081, China

**Keywords:** lotus–fish co-culture, microbiota, alpha diversity, LEfSe analysis, functional annotation

## Abstract

This study employed integrated metagenomic and environmental analyses to elucidate the ecological roles of microbial communities (viruses, archaea, fungi) in lotus–fish co-culture (LFC) systems versus intensive pond culture (IPC), focusing on their impacts on water quality, sediment health, fish gut immunity, and ecosystem sustainability. The LFC system efficiently decreased the concentration of total nitrogen, total phosphorus, chemical oxygen demand in water, and organic matter in sediment via plant–microbe symbiosis. Simultaneously, LFC mediated the structure of microbial communities: archaea enhanced ammonia oxidation and organic matter degradation, which helped counterbalance methanogenesis, whereas the richness of fungi in sediment and fish gut promoted ecological self-purification. The suppression of viral diversity in water helped to reduce pathogen risks. LFC advances sustainable aquaculture by strengthening ecological resilience and health in large-scale operations, achieving dual goals of high productivity and environmental stewardship via closed-loop resource cycling.

## 1. Introduction

Decades-long depletion of wild fish resources, exemplified by the decline in breeding stocks of the “Four Domesticated Fish” in the Yangtze River to below 10% of the historical levels by the 1960s, has driven China to launch the Yangtze River Protection Initiative with its Ten-Year Fishing Ban, spurring aquaculture growth [1,2,3,4]. According to the China Fishery Statistical Yearbook (2020–2024), aquaculture area expanded by 7.56% and total production rose by 16.04%. Rising demand for quality protein and heightened food safety awareness have further accelerated the adoption of eco-friendly and sustainable models in the sector [4,5,6]. Integrated agro-aquaculture systems (e.g., lotus–fish and rice–fish systems) can enhance resource efficiency via dual-use water/land systems [6,7]. It forms a composite system that synergizes ecological restoration, economic valorization, and cultural heritage preservation [8,9]. Lotus–fish co-culture (LFC), practiced in China for millennia, is depicted in the Han Dynasty Jiangnan folk song (Yuefu tradition) as a natural agricultural synergy where fish swim among lotus leaves. Lotus plants absorb excess nutrients from water and soil, providing a natural food source and habitat for fish [10,11,12]. On the other hand, cultured species (e.g., crucian carp and common carp) contribute to pest control, soil decompaction, and waste recycling into organic fertilizers through their excretion [6,10,11,13]. Therefore, the bidirectional synergy in LFC boosts fish health (e.g., growth metrics, immune markers) and optimizes the rearing environment (e.g., water quality, microbiome dynamics), outperforming intensive pond culture (IPC). Unofficial statistics show that China’s lotus pond coverage surpasses 100,000 hectares (ha). Therefore, full-scale implementation of LFC could not only elevate aquatic product quality but also diversify aquatic product portfolios. How can the superiority of the LFC system be quantitatively validated? Microbiome research provides a robust framework to dissect this, leveraging multi-omics integration to uncover mechanistic links.

The microbiome serves as a revolutionary bioindicator for fish health and the aquatic ecosystem, surpassing traditional morphological, physiological, and immunological markers via three strengths: dynamic responsiveness, functional integration, and environmental responsiveness [14,15,16,17]. Acting as an ecological sentinel, it enables early pathogen detection and reveals environmental stress adaptation, making it indispensable for sustainable aquaculture and ecosystem conservation [18,19,20]. Therefore, the microbiome transcends conventional health metrics by integrating host–microbe–environment interactions, offering a paradigm shift in aquatic health monitoring and ecosystem stewardship.

Furthermore, the gut microbiota critically shapes cultured fish health by regulating immune responses, stress response, reproduction, endocrine function, nutrient metabolism, and development [21,22]. Understanding microbial diversity and functional roles has always been pivotal for aquaculture advancement [23,24,25,26]. While current research on aquaculture microbiota predominantly focuses on bacterial communities (with metagenomic analyses providing critical insights into host–pathogen dynamics, disease progression, and ecological interactions) [23,27,28,29,30], bacterial communities engage in complex cross-kingdom interactions with archaea, fungi, and host organisms, collectively modulating host physiology and microbial functionality [28,31]. However, other key microbial taxa—particularly fungi, archaea, and viruses—remain systematically understudied. Major knowledge gaps persist regarding their ecological roles (e.g., nutrient cycling) and pathogenic impacts (e.g., viral outbreaks) in aquaculture systems [25,28].

Mammalian and fish gut microbiotas exhibit conserved functional profiles in nutrient metabolism and immune regulation [32,33]. In addition to bacteria, evolutionarily conserved mechanisms indicate that fungi, archaea, and viruses play critical roles in digestion, endocrine signaling, and stress adaptation [28]. Although archaea typically constitute <2% of gut microbiota (e.g., dominated by Methanomicrobia in freshwater fish), their methanogenesis enhances polysaccharide fermentation and detoxifies bacterial metabolites [34,35,36,37]. Fungi, while similarly low in abundance, degrade nutrients and modulate microbial communities via chitin-rich cell walls, with immunomodulatory functions conserved from teleosts to mammals [38,39,40]. Gut viruses, notably including Herpesviridae and Retroviridae, predominantly maintain immune and metabolic homeostasis through symbiotic interactions, as shown by aquatic virome studies [41,42,43,44].

Current knowledge of archaea, viruses, and fungi in aquaculture remains limited, particularly between IPC and LFC systems. This study bridges the gap via a metagenomic analysis of crucian carp gut and its environment, revealing microbial diversity patterns and ecological impacts across farming systems. By elucidating functional interactions and metabolic contributions, it provides insights on how to optimize microbial-driven nutrient cycling, host resilience, and precision aquaculture.

## 2. Materials and Methods

### 2.1. Experimental Setup

The field experiment was conducted in two adjacent sites of Large-scale Pond Aquaculture (an area of approximately 3.33 ha for both P1 and P2) in Shuangfeng County, Hunan Province, China, with P1 configured as LFC and P2 as IPC. In P1, lotus seedlings were planted directly in the bottom mud, and the recommended planting density for lotus rhizomes is 3.00 m row spacing and 2.00 m plant-to-plant spacing. The P2 implemented traditional pond aquaculture for crucian carp via mud sediment without aquatic vegetation. The fingerlings of crucian carp (95.5 ± 8.3 g, mean ± SD) were introduced into the pond once lotus seedlings reached 0.20–0.30 m in height, with the water level gradually elevated to 0.50 m through controlled hydrological adjustments [11]. Healthy crucian carp fingerlings (20,000 ind/ha) from the National Education Ministry Breeding Center of Polyploidy Fish, Hunan Normal University were stocked in P1 and P2 during May 2022. Both P1 and P2 implemented standardized artificial feeding protocols for crucian carp [45], but P1 received a 50% lower daily feed ration than P2 despite identical feeding frequency and pellet specifications. This reduction was feasible due to the stratified architecture of lotus root systems enhancing benthic–pelagic coupling, which sustains diverse food webs replacing a large part of formulated feeds [10,11]. The two ponds were continuously supplied by rainfall and the nearby Juan River (a tributary of the Xiangjiang River), ensuring stable hydrological conditions for P1 and P2.

### 2.2. Sampling and DNA Extraction

One week after stocking, water samples (four spatial repetitions, Figure 1), sediment samples (four spatial repetitions), and fish gut samples (five to six biological repetitions) were methodically collected for comprehensive monitoring. And the first sampling served as early-stage aquaculture diagnostic samples: E_W1 (water in P1), E_S1 (sediment in P1), E_F1 (fish intestines in P1), E_W2 (water in P2), E_S2 (sediment in P2), and E_F2 (fish intestines in P2). Subsequent to a 7-month aquaculture period, comprehensive sample collection (water, sediment, and fish intestinal contents) was conducted in November as the post-cultivation stage (late-stage) samples, maintaining technical repetitions (4 per pond) and biological repetitions (5–6 individuals) for comparative analysis. And the samples were correspondingly named as L_W1, L_S1, L_F1, L_W2, L_S2, and L_F2, respectively. Furthermore, thirty individual *C. auratus* were randomly sampled from each of P1 and P2, weighed to the nearest 0.1 g, and their mean body weight (±SD) was calculated. All sampling tools were sterilized with 75% ethanol and UV-irradiated. Samples were stored in sterile cryovials pre-treated with RNase/DNase inhibitors. The collected gut contents were rapidly snap-frozen in liquid nitrogen and stored at −80 °C until DNA extraction. Water samples (200 mL) were prefiltered (5 μm) to remove zooplankton and large particles, followed by 0.2 μm Millipore Isopore filtration (Merck KgaA, Burlington, MA, USA) for bacterial collection. Surface sediments (500 g) were harvested using a Peterson mud harvester (TC-600BD-1/40, Institute of Hydrobiology, Wuhan, China) and homogenized, cleaned of visible plant residues, and stored in sterile bags. Both filters and soil samples (2 g) were preserved at −80 °C prior to DNA extraction. Total DNA from intestinal, water, and soil samples were extracted using the E.Z.N.A.^®^ Soil DNA Kit (Omega Bio-tek, Norcross, GA, USA) per the manufacturer’s protocol, quantified using a NanoDrop2000 UV-vis spectrophotometer (Thermo Scientific, Waltham, MA, USA), and stored at −80 °C prior to sequencing.

### 2.3. Physicochemical Parameters in Water and Sediment

In situ measurements of temperature (T, °C), pH, and dissolved oxygen (DO, mg/L) were conducted using an AP-2000 portable multi-parameter analyzer (Shanghai P-NAV Scientific Instruments Co., Ltd., Shanghai, China). Additionally, water and sediment samples were collected for laboratory quantification of physicochemical parameters: total nitrogen (TN), total phosphorus (TP), nitrate nitrogen (NO_3_^−^–N), ammoniacal nitrogen (NH_4_^+^–N), phosphate (PO_4_^3−^–P), nitrite nitrogen (NO_2_^−^–N), and chemical oxygen demand (COD). Standardized methods were employed, including (1) alkaline persulfate digestion UV spectrophotometry (TN, GB 11894-89) [46]; (2) molybdenum–antimony–scandium spectrophotometry (TP, PO_4_^3−^–P, GB 11893-89) [47]; (3) UV spectrophotometry (NO_3_^−^–N, HJ/T 346-2007) [48]; (4) Nessler’s reagent colorimetry (NH_4_^+^–N, GB 7479-87) [49]; (5) UV spectrophotometry (NO_2_^−^–N, GB 7493-1987) [50]; and (6) potassium dichromate digestion (COD, GB/T 32208-2015) [51].

Sediment physicochemical indicators (pH, organic matter, TN, TP, NH_4_^+^–N, NO_3_^−^–N) were analyzed using standardized methods: (1) pH (NY/T 1377-2007) [52]; (2) organic matter (OM) (NY/T 1121.6-2006) [53]; (3) TN (Automatic Kjeldahl, NY/T 1121.24-2012) [54]; (4) TP (NY/T 88-1988) [55]; (5) NH_4_^+^–N (KCl extraction-spectrophotometry, HJ 634-2012) [56]; and (6) NO_3_^−^–N (ultraviolet spectrophotometry, GB/T 32737-2016) [57].

Physicochemical indicator analyses and visualizations were performed using R (v4.2.3) with standardized methods. Statistical comparisons between experimental modes were performed using parametric independent *t*-tests. Welch’s correction was applied when variances were unequal, as determined by Levene’s tests. All statistical procedures aligned with assumptions’ validation. Data were reported as mean ± SD to quantify variability and ensure reproducibility.

### 2.4. Metagenomic Sequencing and Annotation

The metagenomic sequencing and preliminary sequence analysis were conducted by Ma-jorbio Bio-Pharm Technology Co., Ltd. (Shanghai, China). Metagenomic analysis was performed by indexed multiplexed sequencing. Post-sequencing, samples were demultiplexed, and raw FASTQ data underwent quality control using Fastp to remove (1) bases with Phred quality <20; (2) adapter contaminants; (3) reads <50 bp; and (4) host DNA (via BWA alignment) [58]. The high-quality reads were assembled into contigs using Megahit. Open reading frames (ORFs) were predicted from all contigs >300 bp using Prodigal v2.6.3 [59]. Non-redundant gene catalogs were generated by clustering genes at 95% similarity (CD-HIT). The high-quality reads were aligned to the non-redundant gene set using SOAPaligner v2.21 at 95% similarity [60]. Functional annotation was conducted by BLASTP v2.14.0 against the NCBI nonredundant (NR) protein sequences and KEGG database. Gene abundance of each taxonomy or KEGG metabolic potential was calculated as reads per kilobase per million mapped reads (RPKM) to normalize sequencing depth biases [61].

Independent gene catalogues were built for viral, archaeal, and fungal taxa. Non-redundant sequences were annotated using DIAMOND (v2.1.8.162) in BLASTP against NCBI NR (E-value  ≤  1 × 10^−5^) [62], and taxonomic classifications were retrieved from the NCBI taxonomy hierarchy. Species abundances were quantified by summing the gene counts per species. Subsequently, phylogenetic profiling was performed at the eight ranks to generate abundance matrices. Alpha diversity was calculated (mothur, https://mothur.org/ (accessed on 25 December 2024)) and compared using an ANOVA (Tukey’s HSD). Community similarity/dissimilarity was visualized using a PCoA based on Bray–Curtis dissimilarity, and group differences were tested using Adonis, ANOSIM, and PERMANOVA (999 permutations). LEfSe was used to identify ecologically divergent taxa between LFC and IPC. Their discriminatory power was quantified by linear discriminant analysis (LDA) with habitat-specific thresholds: water: LDA > 3.5 (reduce noise); sediment: LDA > 2.5 (balance sensitivity and specificity); and fish gut: LDA > 4.0 (reduce false positives). Top five discriminatory taxa per habitat were visualized.

## 3. Results

### 3.1. The Physicochemical Properties in Water and Sediment

Contrary to expectations, the crucian carp in P1 achieved a comparable growth performance (0.39 ± 0.18 kg/ind) to those in P2 (0.45 ± 0.09 kg/ind). The physicochemical indicators in water and sediment of P1 and P2 were determined using Chinese national standard (GB) methods. In the early-stage aquaculture, all physicochemical indicators showed statistically significant differences between P1 and P2 (Appendix A), while the absolute disparities in measured values remained relatively small (e.g., TN: 2.74 ± 0.22 vs. 2.51 ± 0.13 mg/L, *p* = 0.011; TP: 0.13 ± 0.03 vs. 0.15 ± 0.01 mg/L, *p* = 0.038, Table 1). However, during the later-stage aquaculture, nearly all values of physicochemical indicators demonstrated marked variations between P1 and P2 (e.g., TN: 1.45 ± 25 vs. 2.77 ± 0.14 mg/L, *p* = 3.6 × 10^−5^, Appendix A), with the exception of pH values in both water and sediment remaining similar. In P1, except for NO_3_^−^-N, T, DO, and TP, the other six water quality parameters were all significantly lower in the late stage than early (*p* < 0.01, Appendix A). In contrast, most water quality parameters in P2 exhibited a pronounced upward trend in the late stage, with pH and NO_2_^−^–N being the exception. Although the trends of these physicochemical indicators in the sediment generally paralleled those in the water, the magnitude of their variations was relatively smaller. Notably, there were no significant differences between early and late stages in P1 sediment indicators. However, TN and NH_4_^+^-N in P2 sediment demonstrated a substantial increase in the late stage (*p* < 0.01, Appendix A), with their levels significantly surpassing those in the early stage.

### 3.2. Quality Assessment in Metagenomic Sequencing

Post-QC high-quality sequence counts exhibited a range of 4.6 × 10^7^ to 5.8 × 10^7^ across samples, with a mean of 4.6 × 10^7^ (±0.4 × 10^7^) sequences per sample (Appendix A). The proportion of high-quality sequences to raw sequences averaged 98.1% (range: 95.8–99.0%). Q20 values averaged 97.1% (range: 94.8–98.5%) across samples, while Q30 values averaged 93.8% (range: 92.1–95.5%), and the GC content averaged 53.8% (range: 41.5–61.5%). The metagenomic sequencing data demonstrated high reliability with Q30 > 90%, GC content stability (53.8% ± 4.7%), and validated reproducibility across technical replicates, meeting the quality thresholds for the following genomic analysis.

### 3.3. Alpha Diversity of Fungal–Viral–Archaeal Communities

Microbial α-diversity (Chao, Pielou, Shannon indices) was assessed for viral, archaeal, and fungal communities in water, sediment, and crucian carp gut samples (P1/P2). All samples reached full coverage (index = 1), ensuring complete taxon detection (Table 2, Table 3 and Table 4).

Water viral richness remained persistently lower in P1 than P2, with early-stage evenness (0.16 vs. 0.38) and diversity (1.02 vs. 2.60, Shannon index) significantly reduced in P1 (*p* < 0.05), but late-stage values comparable. Sediment viral diversity showed transient reduction in early-stage P1 (*p* < 0.05), while richness and evenness showed no inter-pond differences at either stage. Gut viral richness was suppressed in late-stage P1 (*p* < 0.05), accompanied by early-stage reductions in evenness and diversity (*p* < 0.05), with no late-stage inter-pond differences (Table 2).

Archaeal richness in water exhibited sustained suppression in P1 compared to P2 across both stages (*p* < 0.05), while evenness and diversity showed no inter-pond differences (Table 3). Archaeal richness in sediment was significantly higher in P1 in the early stage, but late-stage values comparable with P2. Archaeal richness and evenness in the gut were similar between P1 and P2 in the early stage. However, in the late stage, P1 exhibited higher richness (*p* < 0.05) but lower evenness (*p* < 0.05).

Sediment fungal richness was significantly lower in P1 vs. P2 (*p* < 0.05), but it became marginally higher in P1 by the late stage (Table 4). For gut fungi, P1 consistently exceeded P2 (*p* < 0.05), with no temporal shifts within ponds. Early-stage gut fungal evenness and diversity showed no inter-pond differences, whereas both indices were significantly elevated in P1 in the late stage (*p* < 0.05).

### 3.4. Structure of Fungal–Viral–Archaeal Communities

Principal coordinate analysis (PCoA) based on the Bray–Curtis distance for viral species at the taxonomic level in water, sediment, and the gut were conducted, respectively. Bray–Curtis-based PCoA revealed the viral community structure, while PERMANOVA indicated no significant spatial/temporal variations either in water or sediment (Figure 2a,b). However, gut viromes displayed significant pond/temporal variations (PC1/2: 37.69%/18.43%; *p* < 0.05, Figure 2c). Unclassified *Caudoviricetes* dominated the water (80.94–95.64%) and sediment (83.10–90.09%) viromes (Figure 3a). Late-stage differentials included *Lavidaviridae* (P1: 3.77%) vs. *Cyanoviridae* (P2: 1.93%) in water; P2 sediment indicated a succession to unclassified virus dominance (12.17%); and gut *Caudoviricetes* varied substantially (46.52–79.93%), with dynamic subdominant taxa (Figure 3a).

Archaeal communities, visualized via Bray–Curtis PCoA, exhibited no significant spatial or temporal variation in water (PERMANOVA, *p* > 0.05), but it did indicate significant inter-pond differences and temporal variations in sediment and gut (PERMANOVA, *p* < 0.05, Figure 2d–f). The dominant archaeal phyla, including *Euryarchaeota* and Candidatus_Thermoplasmatota, demonstrated significant spatiotemporal variations in abundance (Figure 3b). In the late stages, the water exhibited a notable increase in the prevalence of *Euryarchaeota* and *Nitrososphaerota*. The composition of sediment archaea remained stable throughout. The gut archaea were consistently dominated by *Euryarchaeota* (Figure 3b). At the genus level, the water communities displayed divergence in their relative abundances, with P1 indicating major shifts in the late stages (e.g., unclassified archaea decreased from 81.5% to 24%). The sediment maintained a consistent taxonomic composition, albeit with variable abundances. The dominance in the gut shifted from *Methanobacterium* and *Methanosarcina* in the early stages to pond-specific assemblages in the late stages (Figure 3c).

For fungal communities, PERMANOVA detected substantial differences among ponds, as well as notable temporal variations in the water, sediment, and gut samples (*p* < 0.05, Figure 2g–i). Fungal communities exhibited divergence between ponds, with P1′s water shifting to *Trametes* dominance (13.97% compared to P2′s 1.74%) while sediments maintained compositional similarity (Figure 3d). In the gut, fungi underwent a temporal restructuring—in P1, there was a transition from *Vittaforma* to *Hapsidospora* dominance, whereas in P2, there was a shift from *Anncaliia* to *Penicillium*/*Aspergillus* dominance—likely influenced by host–microbe interactions (Figure 3d).

### 3.5. LEfSe Differential Discriminant Analysis of Fungal–Viral–Archaeal Abundance

LEfSe applies linear discriminant analysis (LDA) to identify taxonomic biomarkers exhibiting significant differential abundance across experimental groups. In terms of viral community dynamics in water, P1 demonstrated early dominance of *Caudovirales*, transitioning to *Lavidaviridae* in its late stages (Figure 4a). Conversely, P2 exhibited early enrichment of *Kyanoviridae*/*Marseilleviridae*, shifting to *Kyanoviridae*/*Herpesvirales* by its late stages. In sediments, P1 was initially enriched with *Crassvirales/Zobelliridae/Steigviridae*, later succeeded by *Caudoviricetes*/*Haloferviridae* (Figure 4a). P2, on the other hand, shifted from early *Marseilleviridae* dominance to unclassified viruses/*Autolykiviridae*. In the fish guts, P1 displayed early enrichment of *Uroviricota* and *Caudoviricetes*, evolving into p__*Preplasmiviricota*, o__*Priklausovirales*, and c__*Maveriviricetes*. Meanwhile, P2 transitioned from early *Marseilleviridae* and *Megaviricetes* to *Uroviricota* and *Caudoviricetes* dominance (Figure 4a).

In the water, the archaeal community experienced changes throughout the different stages of P1 and P2 (Figure 4b). During the early stage of P1, there was an enrichment of unclassified archaea, which later gave way to an enrichment of *Woesearchaeota* in P2. It is important to note that the late stage of P2 significantly increased the ratio of *Woesearchaeota* to *Euryarchaeota* as well as methane-producing archaea (*Methanomicrobia*/*Methanomicrobiales*). In the sediments, the early P1 stage was characterized by the presence of g__*Methanoregula*/p__*Thermoplasmatota*. As it progressed, a shift occurred towards f__*Methanoregulaea*/g__*Methanoregula*. The P2 stage began with a dominance of unclassified archaea/g__*Methanothrix*, which eventually advanced to a dominance of *Methanomicrobiales* (Figure 4b). Within the guts, the P1 stage saw a transition from early *Methanobacteriales* to late-stage *Euryarchaeota* and taxa related to *Methanobacteriales*. In contrast, the P2 stage switched from *Methanobrevibacter* to a dominance of unclassified archaea (Figure 4b).

In the fungal community, P1 in water transitioned from an early dominance of *Ascomycota* to a late-stage supremacy of *Agaricomycetes*, while P2 shifted from *Chytridiomycota*-related taxa to a dominance of *Saccharomycetes* (Figure 4c). In sediments, P1 evolved from an enrichment of *Stachybotrys* to taxa associated with *Spizellomycetales*; and P2 transitioned from early *Chytridiomycota* to *Boletales*/*Dothideomycetes* (Figure 4c). In fish guts, P1 restructured from early *Vittaforma* to late *Dothideomycetes* dominance; and P2 displayed a late-stage enrichment of *Vittaforma* following an early prevalence of *Tubulinosema* (Figure 4c).

### 3.6. Environmental Driver Associations with Fungal–Viral–Archaeal Abundance

Type I error control in multiple correlation analyses was achieved via the Benjamini–Hochberg false discovery rate (FDR) procedure, with the significance threshold set at an FDR < 0.05. We analyzed viral taxa–environment correlations using heatmaps (Figure 5), revealing distinct preferences: *Phycodnaviridae* and algal DNA virus correlated positively with COD, TN, NH_4_^+^-N, NO_2_^−^-N, TP, and PO_4_^3−^-P in water (Figure 5a); *Mimiviridae*, *Autolykiviridae*, and *Guttaviridae* were associated with TN/TP/NH_4_^+^-N (*Autolykiviridae* additionally with NO_2_^−^-N/PO_4_^3−^-P, *Guttaviridae* with COD); *Megaviricetes* was linked with T/DO; while *Marseilleviridae* and *Iridoviridae* showed pH-dependence. In sediment, *Megaviricetes*, *Phycodnaviridae*, *Mimiviridae*, and *Iridoviridae* inversely correlated with OM; *Phycodnaviridae* and *Mimiviridae* decreased with pH; and *Marseilleviridae* covaried positively with NH_4_^+^-N/NO_3_^−^-N (Figure 5a).

In water, the archaea *Euryarchaeota*, *Thermoproteota*, *Woesearchaeota*, and *Nitrososphaerota* correlated positively with NO_3_^−^-N and COD (Figure 5b); *Thermoproteota* additionally showed a negative pH correlation. *Bathyarchaeota*, *Lokiarchaeota*, and *Thermoplasmata* was associated with elevated TN/NH_4_^+^-N/COD/TP, while *Nanoarchaeota* demonstrated broader nutrient linkages (TN/NH_4_^+^-N/COD/TP/PO_4_^3−^-P/NO_2_^−^-N). Most archaeal genera exhibited similar TN/NH_4_^+^-N/TP/COD correlations. In sediment, *Lokiarchaeota* and *Thermoplasmata* increased with TP but decreased with pH/OM; *Euryarchaeota* and *Thermoproteota* similarly declined with pH/OM; and Candidatus *Verstraetearchaeota* was inversely correlated with pH. Genus-level analyses confirmed these patterns: unclassified taxa within *Thermoproteota*, *Thermoplasmata*, *Euryarchaeota*, and *Methanosaetaceae* thrived under high TP but diminished with increasing pH and OM, mirroring the sensitivity of *Methanoregula* and unclassified *Methanoregulaceae* to pH and OM gradients (Figure 5b).

In water, the abundance of the fungal *Trametes* increased in correlation with T and DO, whereas *Coelomomyces* exhibited an inverse relationship with these variables but showed a positive correlation with TN, NH_4_^+^-N, TP, PO_4_^3−^-P, and NO_2_^−^-N (Figure 5c). Distinctively strong nutrient associations were observed for *Rhizophydium*, *Entophlyctis*, *Hyphochytrium*, *Blastocladia*, and *Globomyces* (all *p* < 0.01), with the latter three genera preferring habitats that were cooler and had lower oxygen levels. The populations of *Rhizosporangium* and *Rhizopus* were dependent on pH and sensitive to NO_3_^−^-N, thus exhibiting contrasting distribution patterns along the temperature–oxygen gradient. In sediment, the abundance of *Mucor* increased with TP but decreased with pH and OM. Conversely, *Aspergillus* was negatively correlated with NH_4_^+^-N, *Lipomyces* displayed a pH-dependent distribution, while *Rhizophagus* populations showed a positive correlation with TN (Figure 5c).

### 3.7. Functional Annotation and Statistical Analysis of Fungal–Viral–Archaeal Genes

The key findings from the comparative metagenomic analyses of viral, archaeal, and fungal gene functions across different aquaculture phases (early vs. late) and modes (LFC vs. IPC) are summarized below (Figure 6a–c).

Viral functions displayed a significantly higher relative abundance in water compared to sediment or fish gut microbiomes, with no discernible temporal (early vs. late) or spatial (P1 vs. P2) differences noted in water and sediment, suggesting functional stability. The primary metabolic pathways for water viruses encompassed Global and Overview Maps, Replication and Repair, Metabolism of Cofactors and Vitamins, Nucleotide Metabolism, Carbohydrate Metabolism, and Glycan Biosynthesis and Metabolism. Gut viral communities exhibited minimal functional variation between P1 and P2 in the early stage, but most pathways (e.g., Global Maps, Nucleotide Metabolism, Replication/Repair) were significantly lower in P1 compared to P2 by the late stage (Figure 6a).

The abundances of functional genes in Archaea were notably lower in water compared to sediment and gut environments. Temporally and spatially, the profiles remained consistent between P1 and P2 stages. For early-stage sediment archaeal functions such as Methane Metabolism, Carbon Metabolism, and Propanoate Metabolism, P1 exhibited significantly higher values than P2. However, this disparity was not evident in the late stage. In the gut, while P1 demonstrated reduced core archaeal functions (e.g., Methane Metabolism, TCA Cycle) initially, it eventually surpassed P2 in the late stage (Figure 6b).

The functional profiles of fungi were substantially elevated in water and gut samples compared to sediment, with no discernible P1/P2 differences observed across stages in these compartments. In the late stage, sediment from P1 exhibited a significant upregulation of pathways related to human diseases (e.g., Diabetic Cardiomyopathy, Shigellosis) and cellular functions (e.g., Protein Processing in Endoplasmic Reticulum, Endocytosis). The fungal functions in the gut during P1 were marginally lower than those in P2′s late stage. Temporally, the abundance of water fungi peaked in the early stage and declined in the late stage, while the abundance in the gut increased by the late stage (Figure 6c).

## 4. Discussion

Compared to IPC, the LFC system demonstrated significantly improved water and sediment quality, characterized by lower levels of TN, TP, NO_2_^−^-N, and COD in water and reduced OM and TN in sediment. Furthermore, it exhibited reduced viral risks in both the environment (lower viral richness) and the fish gut (reduced viral functional abundance). Concurrently, the LFC system enhanced beneficial archaeal communities, showing increased diversity and metabolic function abundance in the fish gut. Finally, the LFC displayed an improved fungal community within the fish gut, marked by higher richness and a stable structure. In the following, we present a comparative analysis of both physicochemical parameters and microbial communities between LFC and IPC in detail.

In aquatic ecosystems, macrophytes play a crucial role in water purification by absorbing decomposed nutrients from both the water and sediment, thereby enhancing the remediation of aquaculture water quality [63,64]. Most physicochemical parameters of water and sediment between P1 (LFC) and P2 (IPC) were found to be significantly different in the early stage, but the differences were relatively small in magnitude. In the later stage, P1 showed significantly lower values than P2 for all parameters except pH and sediment TP (Appendix A), suggesting that the lotus played a significant role in reducing nitrogen and phosphorus loads in water, thereby decreasing the accumulation of organic pollutants and contributing to the water purification [65,66]. Additionally, ecological floating beds in rice–aquaculture systems effectively removed N and P; and aquaponics systems significantly reduced NH_4_^+^-N and NO_2_^−^-N concentrations in aquatic environments [19,67,68]. The OM content in P1 was significantly lower than that in P2 (*p* < 0.05). This suggested that LFC effectively enhanced the degradation and utilization of organic matter, thereby reducing its accumulation in sediments, as opposed to IPC [65]. LEfSe revealed enrichment of organic matter-degrading microorganisms in LFC sediments, with fungal (*Trichoderma*, *Aspergillus*) and archaeal (*Methanosarcina*) taxa as key contributors. Tao et al. (2012) quantified OM and TN reduction in lotus ponds, showing 0.59 g/kg OM and 0.74 mg/kg TN loss, respectively [69]. The LFC could significantly reduce the nitrogen content in sediment and even reduce the release of nitrogen and phosphorus [12,66]. Therefore, LFC exhibits substantial benefits over IPC in diminishing nitrogen and phosphorus loads and alleviating organic pollution in aquatic ecosystems.

Notably, viruses, regarded as the most diverse biological entities on our planet, display an unprecedented functional diversity in controlling nutrient cycles across various ecosystems, including freshwater, marine, and terrestrial systems [70,71]. Their ecological integration—from nutrient cycling to microbiome modulation—underscores their indispensable role in planetary biogeochemistry [71,72]. Viral diversity dynamics are influenced by both biotic and abiotic environmental factors [73,74]. This study revealed that the viral richness under the LFC demonstrated a significantly lower diversity compared to the IPC, with notable differences observed in both water and fish intestinal samples (*p* < 0.05, Table 2). The observed disparity likely stemmed from elevated nutrient inputs (e.g., aquafeed) in IPC, which elevated the OM content and created eutrophic conditions that enhance viral proliferation. Conversely, LFC systems mitigated nitrogen/phosphorus loads, reducing sediment OM and nitrogen accumulation (Appendix A). This environmental optimization suppressed viral transmission and replication, consequently decreasing infections in fish intestines [75,76]. The diversity of viruses in freshwater ecosystems is influenced by multiple environmental factors, particularly nutrient levels [77,78]. Higher concentrations of nutrients (e.g., nitrogen, phosphorus) in aquatic systems correlated with elevated viral diversity and species richness, driven by enhanced microbial productivity and host availability [79]. Our results revealed that the vast majority of viruses exhibited significant positive correlations with TN, NH_4_^+^-N, and TP (*p* < 0.05, Figure 5a), suggesting their tight coupling with nutrient cycling in aquatic ecosystems [80,81]. Furthermore, dominant viruses (e.g., *Caudovirales*) reduce viral community alpha diversity by competitively occupying key niches like host receptors or nutrients, driving winner-takes-all dynamics [82,83].

In contrast to IPC, LFC displayed no substantial differences in the dominant viral species and prevalent viral taxa in water and sediment (Figure 3a). However, distinct disparities were observed in the viral community structures within the fish’s intestine, especially in the late stage (Figure 3a). In contrast to LFC, IPC demonstrated a significant enrichment of *Kyanoviridae* and *Herpesviruses* in water (Figure 4a). *Kyanoviridae*, viruses that specifically infect cyanobacteria, have a broad distribution in various environments such as lakes, ponds, and sediments [84]. They showed considerable potential in controlling algal blooms, regulating cyanobacterial populations, and maintaining water quality through mechanisms like host lysis and nutrient cycling [85,86]. *Herpesviruses* are notable pathogens that extensively spread amongst mammals, birds, and fish. They pose a severe risk to the health of fish, particularly during aquaculture [87,88]. Recent research has determined that the gill-bleeding disease observed in farmed crucian carp across China is attributed to herpesvirus infection, which has brought heavy losses to some aquaculture farms in China [89,90]. High levels of two viruses in the IPC pond water indicated algal blooms and worsening water quality, threatening fish health and growth. *Cyanobacteriophages* (algae-targeting viruses) showed strong positive correlations with key pollutants, including COD, TN, NH_4_^−^-N, NO_2_^−^-N, TP, and PO_4_^3−^-P (Figure 5a). This result further highlighted the high risk of water degradation in the IPC, while the LFC could maintain a relatively high-quality water environment by rhizosphere filtration and microbial self-purification, thus reducing viral pathogen transmission risks. The inhibition of herpesvirus mediated by LFC does not merely augment production by diminishing mortality rates; it also transforms disease management paradigms. This is achieved by decreasing reliance on chemical controls and enriching antagonistic microbiota, which competitively inhibit viral proliferation [91].

Furthermore, the fish intestinal viral communities exhibited significantly greater diversity in the LFC group compared to the IPC group (Table 2, Figure 3a and Figure 4a), with a more stable community structure [92]. Functional analysis indicated suppressed virus metabolic activity in the LFC, contrasting with the active metabolism observed in the IPC. This discrepancy may reflect unfavorable ecological conditions in the LFC that suppress viral replication [8], thereby bolstering host immunity and intestinal health.

Notably, ecological regulation of viral dynamics rarely operates in isolation; archaeal groups, often underappreciated, frequently act as critical intermediaries within microbial communities [93]. As crucial drivers of biogeochemical cycles, archaeal groups possess extraordinary genetic and metabolic diversity. Specific archaeal groups act as keystone species, shaping microbial community composition and function [93]. Their implications for host health and disease prevention remain significant yet largely unexplored [3,94,95]. Compared with the IPC, the sediment archaeal richness in LFC exhibited a significant increase in the early stage (*p* < 0.05, Table 3). Archaeal richness and diversity in sediments were strongly influenced by plant root exudates, as evidenced by previous studies [96,97]. Lotus roots act as pivotal ecosystem engineers within sediment systems, exerting their influence through three interconnected mechanisms: (1) physical restructuring that enhances sediment porosity and promotes particle aggregation; (2) chemical regulation that modulates redox potential and mediates nutrient fluxes; and (3) biological facilitation of microbial consortia, which drive organic matter degradation and pollutant transformation [98,99,100]. As a result, during the initial cultivation, lotus roots modified sediment, elevating archaeal richness. Later growth stagnation weakened this effect, leading to a gradual convergence of the archaeal community within the sediment between LFC and IPC (Table 3).

*Nitrososphaerota* are key drivers of nitrification in the nitrogen cycle and critical participants in the carbon cycle [101]. Our results revealed that *Nitrososphaerota* abundance was significantly positively correlated with NO_3_^−^-N levels in water (Figure 5b). By enhancing *Nitrososphaerota*-mediated ammonia oxidation and organic decomposition capacity, LFC promotes efficient nitrogen transformation, reducing toxic NH_4_^+^-N/NO_2_^−^-N accumulation and improving water quality (Appendix A). Substantial research has demonstrated the pivotal role of ammonia-oxidizing archaea (AOA) in wastewater treatment and nitrogen pollution remediation [29,102,103]. *Euryarchaeota* exhibited high relative abundance across diverse habitats under both LFC and IPC systems, representing a pivotal archaeal phylum with a ubiquitous distribution and significant ecological functions [104,105]. Furthermore, unclassified archaea showed a high abundance in sediments and fish gut samples, indicating unresolved ecological and host-adaptive functions that warrant deeper investigation.

Archaeal communities differed significantly between LFC and IPC sediments (Figure 4b). LFC enriched methane-regulating archaea (e.g., *Methanoregulaceae*, *Methanoregula*) that couple organic decomposition with methanogenesis to sustain methane equilibrium [106,107]. In contrast, IPC sediments were dominated by obligate acetoclastic (*Methanothrix*) and versatile methylotrophic methanogens (*Methanosarcinales*), indicating adaptation to high-organic-load methanogenesis [108,109]. The dominance of *Methanothrix* and *Methanosarcina* in acetoclastic methanogenesis—contributing ∼60% to biogenic methane emissions—highlights their ecological resilience under high-organic-load conditions typical of conventional aquaculture [110,111]. IPC fostered benthic methanogen proliferation through continuous organic deposition, ultimately exacerbating methane flux and sediment anoxia, with concomitant risks of eutrophication and microbial dysbiosis. In contrast, the LFC enriched archaeal consortia (e.g., *Methanoregula* spp.) that orchestrated a methane production–consumption equilibrium via coupled organic decomposition and methanotrophy, reflecting its environmentally benign nature and low-emission characteristics [112]. The *Methanoregula*-driven organic matter depletion in LFC sediments mechanistically explained the observed negative correlation (Figure 5b), coupled with reduced organic matter levels in sediments under the LFC (Appendix A). This aligns with the reported function of *Methanoregula* in accelerating organic matter mineralization while suppressing methanogenesis, thereby depleting sediment organic carbon pools [113].

Furthermore, our findings revealed that the fish gut microbiota in the LFC system exhibited continuous and significant enrichment of *Methanobacteriales*-like archaea, correlating with a more stable gut microbial community structure. The persistent dominance of *Methanobacteriales* archaea (e.g., *Methanobrevibacter*) in LFC fish guts likely stabilized microbial networks by consuming excess H_2_ from bacterial fermentation, thereby suppressing hydrogen accumulation-induced dysbiosis [114]. *Methanobacteriales* archaea utilize H_2_/CO_2_ for methanogenesis while oxidizing formate, sustaining low gut H_2_ pressure [115,116,117]. This dual activity boosts microbial activity, prevents formate buildup, and optimizes intestinal health and microbial balance.

Notably, the metabolic functional abundance of diverse archaeal groups—particularly those involved in methanogenesis, nitrogen fixation, and organic matter mineralization—was markedly elevated in LFC relative to IPC in the late stage (Figure 6b). This archaeal functional enhancement likely expedited sediment nutrient cycling, with methanogenic archaea converting organic waste into bioavailable methane/CO_2_ and nitrogen-fixing archaea increasing ammonium availability for primary producers. The collaborative impact of archaea-driven nutrient recycling and host metabolic optimization collectively enhanced the stability of the aquaculture environment. This resulted in a reduction in ammonia-N accumulation and COD in comparison to IPC (Appendix A), ultimately contributing to improved fish health [118]. The KEGG-annotated archaeal metabolic enhancement and carbon fixation align with LFC’s sustainable design: lotus roots secrete oxygen to create redox gradients, promoting archaeal syntropy with fermentative bacteria while suppressing sulfate-reducing competitors. This archaeal functional resilience underscores LFC’s potential in mitigating aquaculture-induced methane emissions [119].

These archaeal-driven biogeochemical shifts further create niches for fungi, which are relatively rare but play essential roles in substrate mineralization, micronutrient supply, and intestinal barrier function [38,120]. The LFC system demonstrated significantly (*p* < 0.05) higher sediment fungal richness and gut microbiota diversity than IPC (Table 4), with improved community stability in sediment and the fish gut (Figure 2h,i and Figure 3c). Our findings align with evidence that root exudates (e.g., sugars, organic acids, and secondary metabolites) selectively recruit specific fungal taxa such as arbuscular mycorrhizal fungi (AMF), thereby modulating rhizosphere fungal diversity [121,122]. Therefore, the LFC system showed a superior performance over IPC in optimizing aquaculture microenvironments (sediment/gut) and facilitating nutrient–energy transfer, offering practical benefits for sustainable production.

The IPC water was dominated by *Geotrichum* and *Coemansia* fungi, with significant enrichment of chytrids and yeast-like taxa. Notably, *Coemansia* (Zygomycota) represents a saprophytic fungal group associated with organic matter decomposition [123,124]. The dominance of saprophytic *Coemansia* and *Geotrichum* suggested active organic matter processing in IPC water [125]. *Coemansia* abundance positively correlated with nitrogen (TN, NH_4_^+^–N, NO_2_^−^–N) and phosphorus (TP, PO_4_^3−^–P) levels (Figure 5c), indicating organic pollution from sources like fish feed. This pattern implies that *Coemansia* thrives under elevated nutrient conditions, making it a potential microbial sentinel for aquaculture-derived organic pollution. Yeasts are recognized organic matter degraders in eutrophic systems [126], aligning with the observed COD accumulation in IPC water—a hallmark of microbial carbon processing under nutrient enrichment.

In contrast, the LFC water exhibited significant enrichment of fungal taxa from *Agaricomycetes* and *Aspergillus* spp. This enrichment facilitated the enhancement of organic matter turnover by means of extracellular enzymes, thereby mitigating eutrophication through the acceleration of the degradation of complex organic compounds [127,128]. Hyphal exudates of *Rhizophagus* spp. chemoattract phosphate-solubilizing bacteria, forming synergistic partnerships that boost P uptake by plants and fungi [129,130]. Therefore, within the LFC, this interaction likely facilitated fungal-mediated phosphorus cycling, decreasing water phosphorus levels and reducing eutrophication risk. Our findings supported this view, with PO_4_^3−^–P levels significantly decreased at the later vs. early stage in LFC (*p* < 0.001). Conversely, IPC showed a reversal in trend (Appendix A). Notably, late-stage TP in LFC remained significantly lower than in IPC (*p* < 0.001; Appendix A).

Furthermore, compared to IPC, the LFC was dominated by taxa specialized in organic degradation in sediment (e.g., *Trichodermaceae*, *Peziza*, *Monoblepharidomycetes*, Figure 3c). Notably, *Trichoderma* species (*Trichodermaceae*) exert dual advantageous functions: (1) promoting nitrogen/phosphorus transformation and reducing organic phosphorus pollution to improve plant nutrient absorption; and (2) inhibiting pathogens by sequestering Fe^3+^ for biological control [131,132]. Therefore, the LFC achieved (1) dynamic phosphorus regulation through microbial interventions (Appendix A); (2) self-purification through optimized organic degradation (Appendix A); and (3) sustainability by minimizing chemical inputs and reducing pathogen loads (Figure 6c).

The fish intestines in LFC were significantly enriched with the fungal taxa *Dothideomycetes*, *Pleosporales*, *Lindneromyces*, and *Nectriaceae* (Figure 4c). These fungi facilitated the decomposition of organic matter and nutrient cycling, significantly contributing to ecological interactions and host adaptation [133,134]. Notably, they played pivotal roles in pathogen suppression, immune modulation, and the enhancement of symbiotic networks and community stability [135]. In contrast, only the fungal genus *Vittaforma* was significantly enriched in the late-stage IPC fish intestines (Figure 4c), reflecting intestinal environmental homogenization and potentially increased pathogen pressure. Conversely, enriched fungal taxa in the LFC enhanced ecological functions and health benefits, underscoring its superiority for ecosystem stability and sustainability.

In conclusion, the significantly higher fungal abundance observed in LFC sediments and crucian carp intestines (vs. IPC; *p* < 0.05) revealed a heightened ecological self-purification capability. This is likely attributed to the fungal-mediated decomposition of organic matter and suppression of pathogens. Such functional augmentation suggested a superior sustainability potential for the LFC system, potentially due to enhanced nutrient cycling efficiency and resilience against environmental disturbances. These findings on LFC enable scalable aquaculture integration, resolving water pollution, disease risk, and sustainability challenges through microbiome-driven mechanisms. Firstly, the efficacy of LFC stems from its promotion of beneficial archaeal and fungal networks, which suppress viral pathogens while mitigating pollutants (total nitrogen, phosphorus, COD). To scale this self-purification capacity, modular systems could integrate layered sediment zones to enhance archaeal ammonia oxidation and fungal organic matter decomposition, alongside controlled aeration to sustain a low hydrogen partial pressure—a condition stabilized by *Methanobacteria*. Biofilters enriched with these functional microbes may further optimize pollutant removal in industrial aquaculture. Secondly, these findings highlight the role of specific microbes (e.g., *Methanobacteria* in intestines, ammonia-oxidizing archaea, and fungi in sediment) in improving host health and environmental resilience. Probiotics co-encapsulating *Methanobacteria* and antifungal strains (e.g., *Saccharomyces*) can be integrated into aquafeeds to deplete intestinal H_2_, suppress facultative anaerobes (e.g., pathogenic *E. coli* or *Aeromonas*), and reduce infection-driven mortality in high-density aquaculture [136]. Thirdly, LFC’s higher ecological self-purification ability and sustainability potential align with growing consumer and regulatory demands for eco-certified aquaculture.

## 5. Conclusions

The present metagenomic analysis revealed a comparison of microbial communities (viruses, archaea, fungi) between LFC and IPC. The LFC increased sustainability through three primary mechanisms: (1) by reducing viral diversity and stabilizing host–virus interactions, thereby creating a healthier environment; (2) by modifying the structure of archaeal communities to enhance environmental remediation, energy efficiency, and host health; and (3) by optimizing fungal communities, thereby improving phosphorus cycling, organic degradation, and pathogen resistance. Taken together, these microbial-driven effects contribute to a more stable and low-risk aquaculture ecosystem. Future studies must resolve archaeal–fungal syntropy in LFC by quantifying metabolite exchange between ammonia-oxidizing archaea and ligninolytic fungi, and its role in regulating methanogenic pathways. Single-cell transcriptomics of *Methanobacteria* should define H_2_-depletion-mediated suppression of facultative anaerobes, while delineating hydrogenase-dependent immunomodulatory networks that mitigate infection.

## Figures and Tables

**Figure 1 biology-14-01092-f001:**
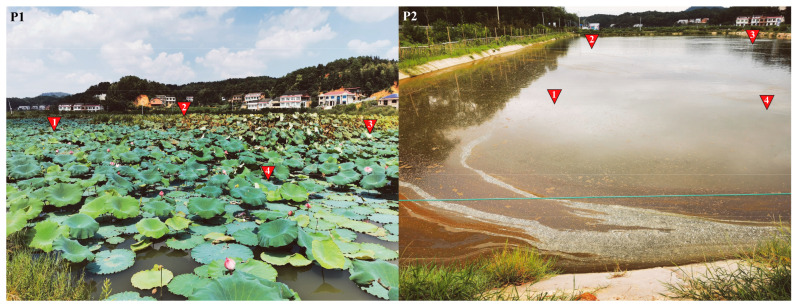
Sampling sites in (**P1**) and (**P2**) (four areas per pond) labeled with red inverted triangles (1–4).

**Figure 2 biology-14-01092-f002:**
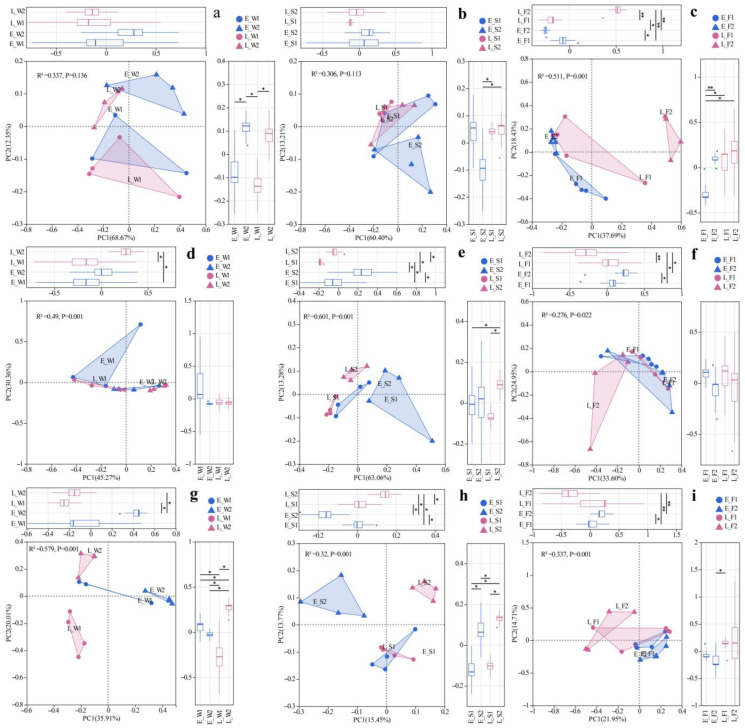
PCoA of viral (**a**–**c**), archaeal (**d**–**f**), and fungal (**g**–**i**) communities across habitats (W: water; S: sediment; F: fish gut) under two models (1: LFC; 2: IPC) and periods (E: early stage; L: late stage). Significance levels: *, *p* < 0.05; **, *p* < 0.01.

**Figure 3 biology-14-01092-f003:**
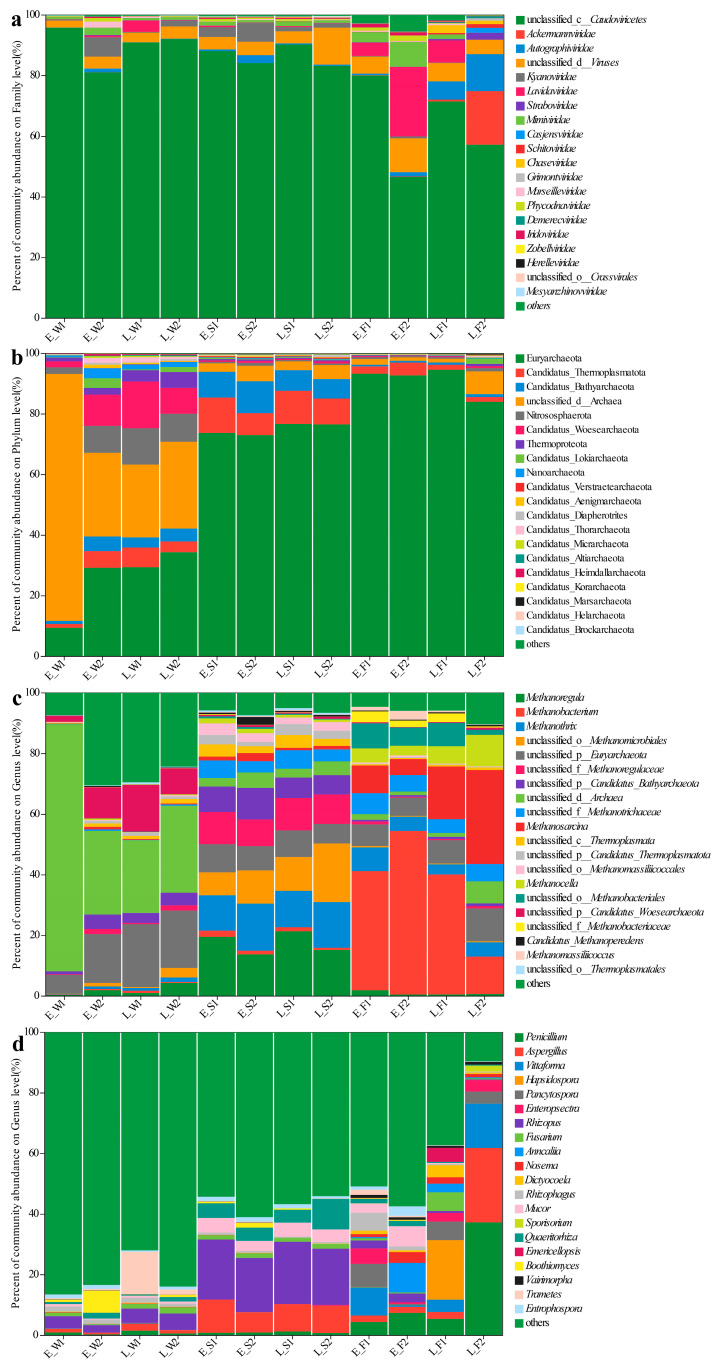
Species composition of viral families (**a**), archaeal phyla (**b**) and archaeal genera (**c**), and fungal genera (**d**) across water (W), sediment (S), and fish gut (F) samples under two culture models (1: LFC and 2: IPC) during early (E) and late (L) culture stages. Different colors represent different taxonomic groups.

**Figure 4 biology-14-01092-f004:**
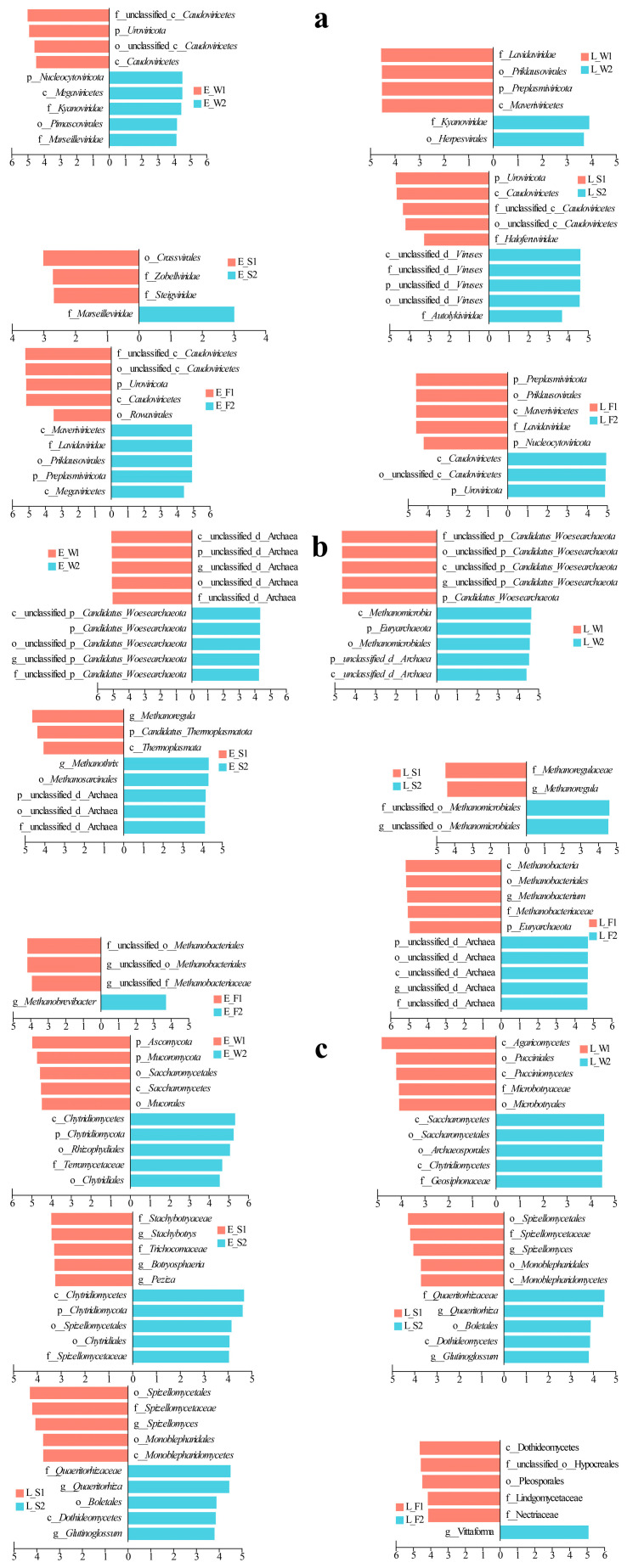
LEfSe analysis of hierarchical viral (**a**), archaeal (**b**), and fungal (**c**) taxa across water, sediment, and fish gut samples under two culture models (LFC and IPC) during early and late culture stages.

**Figure 5 biology-14-01092-f005:**
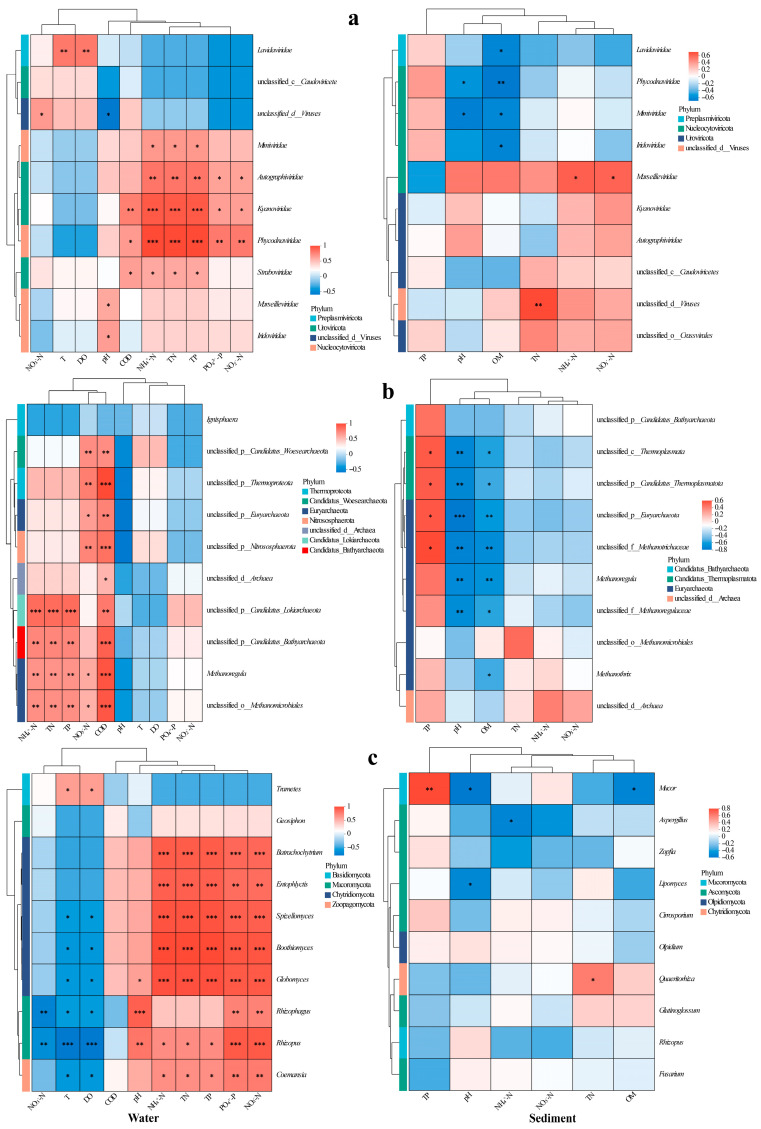
Heatmaps of correlations between major viral families (**a**), archaeal genera (**b**), and fungal genera (**c**) and environmental indicators in water (left) and sediment (right). Significance levels: *, *p* < 0.05; **, *p* < 0.01; ***, *p* < 0.001.

**Figure 6 biology-14-01092-f006:**
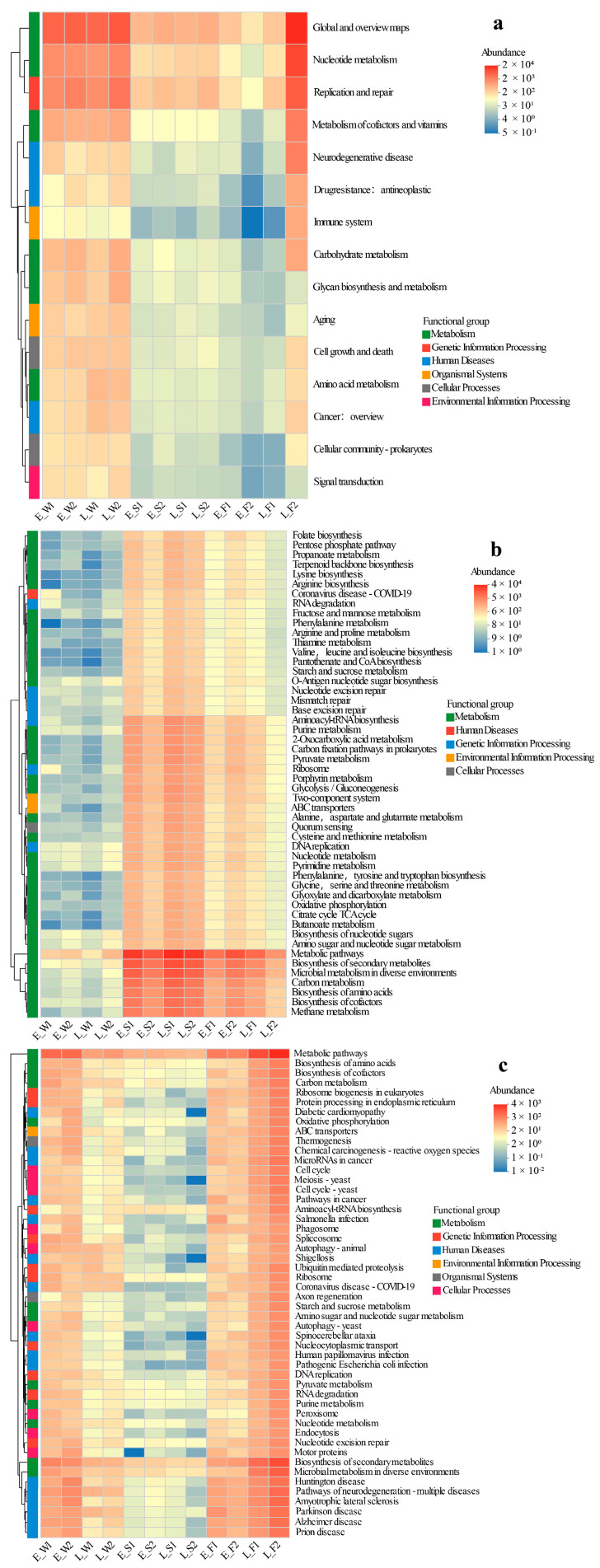
Heatmap clustering of viral KEGG pathways (hierarchy level 2 (**a**)), archaeal KEGG functional profiles (level 3 (**b**)), and fungal KEGG functional profiles (level 3 (**c**)) across habitats under two models (LFC and IPC) during the early and late stages.

**Table 1 biology-14-01092-t001:** Concentrations of physicochemical indicators in water (mg/L) and sediment (mg/kg), mean ± SD.

Samples	T (°C)	pH	DO	TN	TP	PO_4_^3−^–P	NH_4_^+^–N	NO_3_^−^–N	NO_2_^−^–N	COD	OM (g/kg)
E_W1	20.61 ± 0.55	7.56 ± 0.19	3.63 ± 0.15	2.74 ± 0.22	0.13 ± 0.03	0.06 ± 0.01	0.29 ± 0.03	0.13 ± 0.02	0.02 ± 0.00	46.73 ± 3.42	—
E_W2	23.86 ± 0.57	8.45 ± 0.21	3.44 ± 0.23	2.51 ± 0.13	0.15 ± 0.01	0.08 ± 0.00	0.36 ± 0.03	0.27 ± 0.02	0.02 ± 0.00	51.87 ± 2.04	—
L_W1	25.35 ± 0.39	6.88 ± 0.14	3.81 ± 0.30	1.45 ± 0.25	0.11 ± 0.05	0.01 ± 0.00	0.22 ± 0.04	0.43 ± 0.02	0.00 ± 0.00	31.98 ± 2.66	—
L_W2	26.55 ± 0.27	6.76 ± 0.09	3.77 ± 0.78	2.77 ± 0.14	0.44 ± 0.06	0.09 ± 0.00	0.42 ± 0.12	0.83 ± 0.01	0.00 ± 0.00	77.02 ± 6.17	—
E_S1	—	5.60 ± 0.23	—	2.02 ± 0.05	0.63 ± 0.04	—	60.04 ± 6.03	0.94 ± 0.05	—	—	34.29 ± 0.35
E_S2	—	6.00 ± 0.12	—	2.12 ± 0.09	0.55 ± 0.03	—	54.10 ± 1.56	1.16 ± 0.12	—	—	38.92 ± 2.19
L_S1	—	5.60 ± 0.15	—	2.02 ± 0.07	0.61 ± 0.04	—	57.44 ± 3.69	0.96 ± 0.09	—	—	34.40 ± 0.16
L_S2	—	5.30 ± 1.49	—	2.34 ± 0.08	0.56 ± 0.03	—	71.88 ± 22.97	1.32 ± 0.32	—	—	40.41 ± 2.40

**Table 2 biology-14-01092-t002:** α diversity indices of viruses in different ponds in different periods (E represents the early stage, L represents the late stage; 1 represents P1, 2 represents P2; W represents water, S represents sediment; F represents fish gut), the same as below.

Sample	Chao Index	Pielou Index	Shannon Index	Coverage
E_W1	700.00 ^b^	0.16 ^b^	1.02 ^b^	1.00
E_W2	1043.75 ^a^	0.38 ^a^	2.60 ^a^	1.00
L_W1	718.75 ^b^	0.23 ^b^	1.49 ^b^	1.00
L_W2	1092.75 ^a^	0.22 ^b^	1.53 ^b^	1.00
E_S1	354.00 ^a^	0.29 ^ab^	1.67 ^b^	1.00
E_S2	425.00 ^a^	0.36 ^a^	2.17 ^a^	1.00
L_S1	447.50 ^a^	0.24 ^b^	1.44 ^b^	1.00
L_S2	475.00 ^a^	0.24 ^b^	1.93 ^ab^	1.00
E_F1	482.40 ^b^	0.41 ^b^	2.53 ^b^	1.00
E_F2	436.00 ^b^	0.61 ^a^	3.69 ^a^	1.00
L_F1	444.20 ^b^	0.58 ^a^	3.47 ^a^	1.00
L_F2	1137.00 ^a^	0.53 ^ab^	3.68 ^a^	1.00

Different lowercase letters indicate significant differences (*p* < 0.05), the same as below.

**Table 3 biology-14-01092-t003:** α diversity indices of archaea in different ponds in different periods.

Sample	Chao Index	Pielou Index	Shannon Index	Coverage
E_W1	193.67 ^b^	0.41 ^a^	2.19 ^a^	1.00
E_W2	259.00 ^a^	0.58 ^a^	3.4 ^a^	1.00
L_W1	196.25 ^b^	0.60 ^a^	3.17 ^a^	1.00
L_W2	373.75 ^a^	0.52 ^a^	3.09 ^a^	1.00
E_S1	613.00 ^a^	0.50 ^ab^	3.22 ^a^	1.00
E_S2	554.25 ^b^	0.54 ^a^	3.39 ^a^	1.00
L_S1	650.00 ^a^	0.48 ^b^	3.08 ^a^	1.00
L_S2	640.00 ^a^	0.47 ^b^	3.03 ^a^	1.00
E_F1	483.20 ^a^	0.54 ^b^	3.31 ^a^	1.00
E_F2	559.40 ^a^	0.52 ^b^	3.26 ^a^	1.00
L_F1	485.40 ^a^	0.55 ^b^	3.38 ^a^	1.00
L_F2	349.00 ^b^	0.62 ^a^	3.58 ^a^	1.00

Different lowercase letters indicate significant differences (*p* < 0.05).

**Table 4 biology-14-01092-t004:** α diversity indices of fungi in different ponds in different periods.

Sample	Chao Index	Pielou Index	Shannon Index	Coverage
E_W1	600.33 ^ab^	0.83 ^a^	5.24 ^a^	1.00
E_W2	756.00 ^a^	0.79 ^a^	5.25 ^a^	1.00
L_W1	392.75 ^bc^	0.78 ^a^	4.63 ^a^	1.00
L_W2	501.75 ^b^	0.84 ^a^	5.19 ^a^	1.00
E_S1	142.25 ^b^	0.83 ^a^	4.12 ^a^	1.00
E_S2	217.75 ^a^	0.85 ^a^	4.56 ^a^	1.00
L_S1	167.50 ^ab^	0.84 ^a^	4.28 ^a^	1.00
L_S2	146.75 ^b^	0.82 ^a^	4.08 ^a^	1.00
E_F1	690.00 ^a^	0.73 ^a^	4.78 ^a^	1.00
E_F2	554.60 ^a^	0.69 ^a^	4.35 ^a^	1.00
L_F1	669.00 ^a^	0.71 ^a^	4.57 ^a^	1.00
L_F2	502.50 ^a^	0.42 ^b^	2.43 ^b^	1.00

Different lowercase letters indicate significant differences (*p* < 0.05).

## Data Availability

The data are contained within the article.

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
