# Peer review of "Microbiome Diversity and Dynamics in Lotus–Fish Co-Culture Versus Intensive Pond Systems: Implications for Sustainable Aquaculture"

_biology, 2025, doi:10.3390/biology14081092_

Round 1
Reviewer 1 Report
Comments and Suggestions for Authors
This study compares multi-kingdom microbial communities (viruses, archaea, fungi) in lotus-fish co-culture (LFC) and intensive pond culture (IPC) systems using metagenomic and environmental analyses. It reveals microbial mechanisms underlying LFC's advantages in environmental purification, host immunity, and sustainability. The well-designed study provides robust data and valuable insights for sustainable aquaculture development. The manuscript is logically structured but would benefit from some refinements to strengthen scientific rigor.
​1. Terminology Consistency:
Abbreviations (e.g., "OM") should be spelled out at first use ("organic matter").
In the abstract, "augmenting" could be replaced with "enhancing" for better academic phrasing.
​2. Data Presentation and Statistics:
Key environmental parameters (e.g., TN, TP) should include specific value ranges and standard deviations to more clearly illustrate differences between LFC and IPC.
The statement that viral diversity is "suppressed" while intestinal viral communities exhibit greater stability warrants further elaboration. Providing specific diversity indices (e.g., Shannon index) and discussing the apparent contradiction between "suppression" and "stability" would strengthen the argument.
​3. Methodological Clarification:
The LEfSe analysis applied different LDA thresholds for distinct habitats (water: 3.5, sediment: 2.5, intestine: 4.0), but the rationale for selecting these thresholds was not explained. A justification for these values should be added to the "Materials and Methods" section.
This study holds significant scientific merit and is recommended for acceptance after addressing the above points. With these revisions, the mechanistic explanations will gain further completeness, and the work could serve as a paradigm for future research in aquaculture microbial ecology.
Author Response
Reviewer 1
Comments and Suggestions for Authors
This study compares multi-kingdom microbial communities (viruses, archaea, fungi) in lotus-fish co-culture (LFC) and intensive pond culture (IPC) systems using metagenomic and environmental analyses. It reveals microbial mechanisms underlying LFC's advantages in environmental purification, host immunity, and sustainability. The well-designed study provides robust data and valuable insights for sustainable aquaculture development. The manuscript is logically structured but would benefit from some refinements to strengthen scientific rigor.
Response: We sincerely appreciate your insightful evaluation and constructive feedback. We are particularly gratified by your recognition of our study on comparing microbial communities and its contribution to understanding the ecological mechanisms underlying lotus-fish co-culture systems, with important implications for sustainable aquaculture.
​1. Terminology Consistency:
Abbreviations (e.g., "OM") should be spelled out at first use ("organic matter").
In the abstract, "augmenting" could be replaced with "enhancing" for better academic phrasing.
Response 1: Thanks for your suggestions, the “OM” has been spelled out "organic matter" in the abstract. And the "augmenting" has been changed as "enhancing" in the abstract.
​2. Data Presentation and Statistics:
Key environmental parameters (e.g., TN, TP) should include specific value ranges and standard deviations to more clearly illustrate differences between LFC and IPC.
The statement that viral diversity is "suppressed" while intestinal viral communities exhibit greater stability warrants further elaboration. Providing specific diversity indices (e.g., Shannon index) and discussing the apparent contradiction between "suppression" and "stability" would strengthen the argument.
Response 2: Thanks for your constructive suggestion, a new table (Table 1) presented the environmental parameters in the revised manuscript.
Table 2 (Results section) presents the Shannon index and other diversity metrics, demonstrating significantly lower viral α-diversity in the LFC system (highlighted in yellow). The relevant data are presented in the Results section (highlighted in yellow, Lines 261)
​"Viral diversity suppression" characterizes the LFC system's aquatic and sediment compartments, whereas "stability" pertains specifically to the fish intestinal virome. These represent distinct but non-conflicting ecological dynamics. In the Discussion, we elaborate how dominant viral taxa (e.g., Caudovirales) occupy key ecological niches (e.g., host receptors/resources), suppressing colonization by competing viral groups (highlighted in yellow in the revised manuscript, Line 488-491).
​3. Methodological Clarification:
The LEfSe analysis applied different LDA thresholds for distinct habitats (water: 3.5, sediment: 2.5, intestine: 4.0), but the rationale for selecting these thresholds was not explained. A justification for these values should be added to the "Materials and Methods" section.
Response 3: Thanks for your constructive suggestion. The explanations for these values have been added in the Materials and Methods (highlighted in yellow in the revised manuscript, Lines 214-216).
This study holds significant scientific merit and is recommended for acceptance after addressing the above points. With these revisions, the mechanistic explanations will gain further completeness, and the work could serve as a paradigm for future research in aquaculture microbial ecology.

Reviewer 2 Report
Comments and Suggestions for Authors
This study provides valuable insight into the microbiome dynamics observed in Lotus-Fish Co-culture compared to intensive pond culture, with implications for sustainable aquaculture. However, minor aspects could be improved:
- The title should be more concise and precise, avoiding the term “Multi-Kingdom”, which gives a general overview. I suggest “Diversity and Dynamics of the Microbiome Contrasting Lotus-Fish Co-Culture and Intensive Pond Culture: Implications for Sustainable Aquaculture.
- Abstract: The use of “multi-kingdom microbiota” could be more specific
“This study compared the microbiota (viruses, archaea, fungi) in water, sediment, and fish (crucian carp) gut between LFC and Intensive Pond Culture (IPC) systems using integrated metagenomic and environmental analyses.”
“The abundance of fungi in sediment and crucian carp intestine in LFC was significantly higher than that in IPC (p < 0.05), showing higher ecological self-purification ability and sustainability potential in LFC. “: It should be emphasized in the conclusion or discussion t highlight the ecological implications of the study, as its mention in the abstract remains t vague.
3. Keywords: The current keywords are insufficient and do not reflect the abstract.
4. Introduction: The introduction is clear and engaging, especially in linking traditional practices/ Chinese culture with scientific insights. However, the transition between paragraphs lines 61-63 lacks fluidity. Similarly, lines 74-76 feel disconnected and should be better integrated. The section from lines 82-104 is overly detailed for an introduction and should be shortened to main focus on the study’s main objectives.
5. MM:
2.1. Experimental setup: please Replace “to optimize habitat stratification” with a reference.
“Both P1 and P2 implement standardized artificial feeding protocols for crucian carp, with P1 receiving 50% lower daily feed ration compared to P2 while maintaining identical feeding frequency and pellet specifications” This should be briefly explained, why 50%?
2.2. Sampling and DNA extraction: It should be "repetitions" and not "replicates."
2.3. Physicochemical parameters in water and sediment: Some sections could be streamlined and shortened. The description of hydrochemical parameter measurement methods is detailed but could be made more readable by breaking it into shorter sections. Adding relevant references to support the methodology would also be beneficial.
2.4 Metagenomic sequencing and annotation: This section should be shortened and made more concise.
6. Results:
“Compared to P2, P1 significantly improved aquatic and benthic environments while reducing viral richness and enhancing archaeal and fungal richness. This effectively suppressed viral diversity expansion and optimized archaeal-fungal community structures across habitats.”: It should be in discussion.
7. Discussion: Upon reviewing each section of the discussion, it is evident that this study presents valuable and insightful findings. However, the current structure, with fragmented paragraphs, limits the flow and coherence of the narrative. For a clearer presentation, the discussion should be reorganized into a more unified structure that logically connects the various points. I recommend revisiting the organization of the section to ensure a smoother transition between ideas, avoiding unnecessary subheadings that disrupt the logical flow. This approach will enhance clarity, eliminate redundancies, and ensure that the content remains precise and scientifically coherent.
Overall, this work is highly relevant and of significant importance. However, minor revisions and structural adjustments are necessary to improve the clarity and coherence of the manuscript.
Comments on the Quality of English LanguageThe English could be improved to more clearly express the research.
Author Response
Reviewer 2
Comments and Suggestions for Authors
This study provides valuable insight into the microbiome dynamics observed in Lotus-Fish Co-culture compared to intensive pond culture, with implications for sustainable aquaculture. However, minor aspects could be improved:
- The title should be more concise and precise, avoiding the term “Multi-Kingdom”, which gives a general overview. I suggest “Diversity and Dynamics of the Microbiome Contrasting Lotus-Fish Co-Culture and Intensive Pond Culture: Implications for Sustainable Aquaculture.
Response 1: Thanks for your constructive suggestion very much, the “Title” has been changed as “Microbiome Diversity and Dynamics in Lotus-Fish Co-Culture versus Intensive Pond Systems: Implications for Sustainable Aquaculture” in the revised manuscript.
- Abstract: The use of “multi-kingdom microbiota” could be more specific
“This study compared the microbiota (viruses, archaea, fungi) in water, sediment, and fish (crucian carp) gut between LFC and Intensive Pond Culture (IPC) systems using integrated metagenomic and environmental analyses.”
“The abundance of fungi in sediment and crucian carp intestine in LFC was significantly higher than that in IPC (p < 0.05), showing higher ecological self-purification ability and sustainability potential in LFC. “: It should be emphasized in the conclusion or discussion highlight the ecological implications of the study, as its mention in the abstract remains vague.
Response 2: Your proposed revisions exhibit greater precision. The “multi-kingdom microbiota” has been changed as your suggestion “microbiota” in the abstract. We sincerely appreciate your suggestion to enhance the explanation regarding the link between fungal abundance and ecological self-purification in LFC systems. As requested, we have substantially expanded the discussion in Lines 648-653 (last paragraph, highlighted in yellow).
- Keywords: The current keywords are insufficient and do not reflect the abstract.
Response 3: As suggested, we have revised the manuscript's keywords to: ​lotus-fish co-culture; microbiota; alpha diversity; LEfSe analysis; functional annotation.
- Introduction: The introduction is clear and engaging, especially in linking traditional practices/ Chinese culture with scientific insights. However, the transition between paragraphs lines 61-63 lacks fluidity. Similarly, lines 74-76 feel disconnected and should be better integrated. The section from lines 82-104 is overly detailed for an introduction and should be shortened to main focus on the study’s main objectives.
Response 4: We appreciate your constructive suggestions. Following your advice, we have added specific statements to improve the fluency of lines 74–76 and enhanced the coherence and overall consistency surrounding lines 89–97. With specific attention to lines 98–108, we summarized existing research. These revisions are highlighted in yellow in the Introduction section.​
- MM:
2.1. Experimental setup: please Replace “to optimize habitat stratification” with a reference.
“Both P1 and P2 implement standardized artificial feeding protocols for crucian carp, with P1 receiving 50% lower daily feed ration compared to P2 while maintaining identical feeding frequency and pellet specifications” This should be briefly explained, why 50%?
Response 5: Thanks for your suggestion, we have replaced “to optimize habitat stratification” with a reference [11], please see the revisions highlighted in yellow.
We appreciate the reviewer's insightful comments regarding the feed reduction in our LFC. In the revised manuscript, we have provided additional clarification (see Lines 128-133) to substantiate the 50% reduction in formulated feed inputs.
2.2. Sampling and DNA extraction: It should be "repetitions" and not "replicates."
Response: Changed accordingly.
2.3. Physicochemical parameters in water and sediment: Some sections could be streamlined and shortened. The description of hydrochemical parameter measurement methods is detailed but could be made more readable by breaking it into shorter sections. Adding relevant references to support the methodology would also be beneficial.
Response: The methods section for water and sediment physicochemical analysis has been revised per your suggestion, with improved clarity and conciseness (Lines 166-181). All critical methodological details have been preserved to ensure study reproducibility.
2.4 Metagenomic sequencing and annotation: This section should be shortened and made more concise.
Response: Following your suggestion, we have carefully revised the 'Metagenomic sequencing and annotation' section to enhance its conciseness. The updated version (lines 190-217) preserves all critical technical parameters while presenting the methodology more succinctly.
- Results:
“Compared to P2, P1 significantly improved aquatic and benthic environments while reducing viral richness and enhancing archaeal and fungal richness. This effectively suppressed viral diversity expansion and optimized archaeal-fungal community structures across habitats.”: It should be in discussion.
Response 6: Thanks for your suggestion, we deleted this section in the revised MS.
- Discussion: Upon reviewing each section of the discussion, it is evident that this study presents valuable and insightful findings. However, the current structure, with fragmented paragraphs, limits the flow and coherence of the narrative. For a clearer presentation, the discussion should be reorganized into a more unified structure that logically connects the various points. I recommend revisiting the organization of the section to ensure a smoother transition between ideas, avoiding unnecessary subheadings that disrupt the logical flow. This approach will enhance clarity, eliminate redundancies, and ensure that the content remains precise and scientifically coherent.
Response 7: We greatly appreciate your careful review and constructive suggestions, we fully agree with your observation that the original structure of the Discussion section, with fragmented paragraphs, hindered the narrative flow and coherence. Following your recommendation, we have revised the Discussion section to enhance its logical organization and unity. Please see Lines 446-447, Lines 469-471, Lines 508-515, Lines 519-524, Lines 593-595.
Overall, this work is highly relevant and of significant importance. However, minor revisions and structural adjustments are necessary to improve the clarity and coherence of the manuscript.
Response: Thank you sincerely for your positive assessment of our work and for recognizing its relevance and significance. We greatly appreciate your thoughtful feedback, which is invaluable for enhancing the quality of our manuscript.

Reviewer 3 Report
Comments and Suggestions for Authors
Review for the paper “Multi-Kingdom Microbiome Dynamics Contrasting Lotus-Fish Co-Culture and Intensive Pond Culture: Implications for Sustainable Aquaculture” by Qian Qian Zeng and co-authors submitted to “Biology”.
The authors of this research paper conducted an analysis of the microbial communities within a lotus-fish co-culture system, comparing it to an intensive pond culture system to understand the ecological mechanisms at play, focusing on the functional contributions of fungi, archaea, and viruses. They found that the LFC system notably improved water quality by significantly reducing various pollutants and decreasing organic matter in the sediment.
The results of this study may have important implications for the design and implementation of sustainable aquaculture systems. The insights gained into the microbial networks within the LFC system, particularly the optimized interactions between archaea and fungi, suggest that modulating these communities can strengthen host immunity in the fish and improve the overall environmental resilience of the aquaculture setting.
Recommendations.
A simple summary section should be included as suggested in the Instructions for Authors.
Abstract.
L 25-27. The authors should provide the full terminology corresponding to the abbreviations (TN, COD TN, TP, NO2–N, COD) upon their first occurrence in the text, or alternatively, consider removing these abbreviations.
L 27, 31, and 36. It is recommended to omit the reporting of p-values within the Abstract section, as this level of statistical detail is conventionally reserved for the main body of the manuscript..
Introduction.
L 45. The authors should clarify the specific factors contributing to this depletion. It would be useful to include statistical data to contextualize the scale of the issue.
L 46-47. They should report how much aquaculture production has increased since the implementation of the ban.
L 58. They should expound on what specific metrics or indicators demonstrate improved fish health.
Material and Methods.
L 114. To enhance geographical clarity and context for readers, the authors are advised to present the locations of all mentioned sampling sites within a clearly labeled map figure.
L 126. As the authors describe the implementation of standardized artificial feeding protocols without providing supporting references, it is essential that they either cite established methodological sources or incorporate sufficient descriptive detail regarding these protocols directly within the main text.
L 133-141. The manuscript designates sample collection at two distinct temporal phases (early-stage and post-cultivation). The authors should define the specific temporal criteria or developmental milestones used to determine these time points.
L 143-153. While detailing the procedures for sample collection, storage, and DNA extraction, the authors have not addressed critical quality control measures. To validate the integrity of the genomic data and the reliability of subsequent results, they must comprehensively report the specific protocols implemented to prevent, detect, and mitigate potential contamination throughout all stages of sample handling and nucleic acid isolation.
L 180. The authors have erroneously classified independent t-tests as non-parametric statistical methods. This characterization is incorrect; independent t-tests are inherently parametric procedures, relying on assumptions of normality and homogeneity of variance. The text should be revised.
L 227-228. The authors should provide a clear justification for the selection of the specific Linear Discriminant Analysis effect size thresholds applied.
Results.
L 232. The authors assert that growth performance indices demonstrate comparable values between groups. To substantiate this claim, it is essential to clarify whether formal statistical comparisons were conducted on these indices.
Supplementary Figures S1 and S2 contain critical data; however, their current presentation is compromised by excessively reduced font sizes.
Section 3.4. When describing statistically significant differences in microbial communities, the authors attribute this interpretation to PCoA results. This constitutes a methodological inaccuracy, as PCoA serves solely as a dimensionality reduction and visualization technique without inherent statistical testing capabilities. The text should be revised to reference PERMANOVA for significance testing.
Figures 1, 3, 4, and 5. The authors should increase the font size.
Section 3.6. In presenting multiple correlation analyses, the authors have not indicated whether statistical adjustments were implemented to control for Type I error inflation inherent in simultaneous hypothesis testing. It is methodologically imperative to specify which multiplicity correction technique (e.g., Bonferroni, Benjamini-Hochberg false discovery rate) was applied to the correlation matrices.
Discussion.
L 481. The authors should report the specific mechanisms or microbial taxa responsible for enhanced organic matter degradation in LFC systems.
L 506. The authors should specify which nutrient thresholds TN, TP) are considered critical for driving viral diversity in freshwater systems.
L 528. The authors should discuss the practical implications of suppressed Herpesvirus activity in LFC fish populations, especially concerning aquaculture productivity and disease management.
L 546. The authors should elaborate on the specific mechanisms by which lotus roots influenced sediment properties.
The authors should discuss how their findings can be scaled or implemented in larger, real-world aquaculture operations.
Comments on the Quality of English LanguageEnglish revisions are required.
Author Response
Comments and Suggestions for Authors
Review for the paper “Multi-Kingdom Microbiome Dynamics Contrasting Lotus-Fish Co-Culture and Intensive Pond Culture: Implications for Sustainable Aquaculture” by Qian Qian Zeng and co-authors submitted to “Biology”.
The authors of this research paper conducted an analysis of the microbial communities within a lotus-fish co-culture system, comparing it to an intensive pond culture system to understand the ecological mechanisms at play, focusing on the functional contributions of fungi, archaea, and viruses. They found that the LFC system notably improved water quality by significantly reducing various pollutants and decreasing organic matter in the sediment.
The results of this study may have important implications for the design and implementation of sustainable aquaculture systems. The insights gained into the microbial networks within the LFC system, particularly the optimized interactions between archaea and fungi, suggest that modulating these communities can strengthen host immunity in the fish and improve the overall environmental resilience of the aquaculture setting.
Response: Thank you for recognizing our analysis of microbial communities in the LFC system, particularly the comparative focus on IPC systems and the functional roles of fungi, archaea, and viruses. You accurately summarized our key findings: the LFC system improves water quality by reducing pollutants and sediment organic matter, and the underlying ecological mechanisms.
Recommendations.
A simple summary section should be included as suggested in the Instructions for Authors.
Response: Thank you for your valuable suggestion. We have added a simple summary section in the revised MS, please see Lines 13-25. The summary is kept concise (≤200 words) to ensure clarity and accessibility, aligning with the journal’s requirements. We appreciate your guidance in strengthening the manuscript’s readability.
Abstract.
L 25-27. The authors should provide the full terminology corresponding to the abbreviations (TN, COD, TN, TP, NO2–N, COD) upon their first occurrence in the text, or alternatively, consider removing these abbreviations.
Response: Changed accordingly. Please see Lines 32-34.
L 27, 31, and 36. It is recommended to omit the reporting of p-values within the Abstract section, as this level of statistical detail is conventionally reserved for the main body of the manuscript.
Response: Changed accordingly.
Introduction.
L 45. The authors should clarify the specific factors contributing to this depletion. It would be useful to include statistical data to contextualize the scale of the issue.
Response: Thank you for your valuable suggestion, we have supplemented the statistical data in the revised MS, please see Lines 53-55.
L 46-47. They should report how much aquaculture production has increased since the implementation of the ban.
Response: Thank you for this important suggestion. We have supplemented specific data on aquaculture production growth following the ban in the revised manuscript (see Lines 56-58).
L 58. They should expound on what specific metrics or indicators demonstrate improved fish health.
Response: Thank you for this insightful suggestion. We have supplemented specific metrics and indicators that demonstrate improved fish health in the revised manuscript (see Lines 69-71).
Material and Methods.
L 114. To enhance geographical clarity and context for readers, the authors are advised to present the locations of all mentioned sampling sites within a clearly labeled map figure.
Response: Thank you for this constructive suggestion. To enhance geographical clarity, we have added a clearly labeled map figure (Figure 1) in the revised manuscript.
L 126. As the authors describe the implementation of standardized artificial feeding protocols without providing supporting references, it is essential that they either cite established methodological sources or incorporate sufficient descriptive detail regarding these protocols directly within the main text.
Response: Thank you for this valuable observation. In the revised MS, we have supplemented a reference [45] to support the detailed descriptions of the standardized artificial feeding protocols specifically applied to Carassius auratus.
L 133-141. The manuscript designates sample collection at two distinct temporal phases (early-stage and post-cultivation). The authors should define the specific temporal criteria or developmental milestones used to determine these time points.
Response: Thank you for this insightful suggestion. We appreciate the opportunity to clarify the temporal criteria for sample collection. We have explicitly defined the two sampling phases based on the actual breeding process: the "early-stage" refers to one week after fry stocking, and the "post-cultivation" phase corresponds to 1–2 weeks before harvest. As there is no universally standardized division for such sampling phases in aquaculture research, the temporal criteria adopted in this study are consistent with common practices in similar ecological investigations of pond ecosystems, where key stages are typically defined according to critical operational milestones (e.g., initial stocking and pre-harvest periods).
L 143-153. While detailing the procedures for sample collection, storage, and DNA extraction, the authors have not addressed critical quality control measures. To validate the integrity of the genomic data and the reliability of subsequent results, they must comprehensively report the specific protocols implemented to prevent, detect, and mitigate potential contamination throughout all stages of sample handling and nucleic acid isolation.
Response: Thank you for this important observation. We have revised the manuscript to streamline the detailed procedures while addressing critical quality control measures. To ensure data reliability, we have supplemented key contamination prevention measures implemented during sample collection and storage: all sampling tools were sterilized with 75% ethanol and UV-irradiated; samples were stored in sterile cryovials pre-treated with RNase/DNase inhibitors. These details are specified in Lines 150-153 of the revised manuscript.
L 180. The authors have erroneously classified independent t-tests as non-parametric statistical methods. This characterization is incorrect; independent t-tests are inherently parametric procedures, relying on assumptions of normality and homogeneity of variance. The text should be revised.
Response: Thank you for pointing out this error. The term "non-parametric" was erroneously attached to t-tests. We have revised the description to specify that parametric independent t-tests with Welch's correction were applied when variances were unequal (per Levene’s tests), and all statistical procedures align with assumptions validation (Lines 184-186).
L 227-228. The authors should provide a clear justification for the selection of the specific Linear Discriminant Analysis effect size thresholds applied.
Response: Response: Thank you for this valuable suggestion. In the revised manuscript, we have added a clear justification for the Linear Discriminant Analysis (LDA) effect size thresholds, please see Lines 215-217.
Results.
L 232. The authors assert that growth performance indices demonstrate comparable values between groups. To substantiate this claim, it is essential to clarify whether formal statistical comparisons were conducted on these indices.
Response: We thank the reviewer for this comment. Formal statistical comparisons of growth performance indices (including fish weight, as detailed in the Materials and Methods section, Lines 150-151) confirmed group comparability at baseline.
Supplementary Figures S1 and S2 contain critical data; however, their current presentation is compromised by excessively reduced font sizes.
Response: Thank you for your observation regarding the font size in Supplementary Figures S1 and S2. The original figures submitted are high-resolution, ensuring clarity even when significantly enlarged. The vague and unclear picture in the reviewed version may have resulted from format conversion during processing. To address this, we will re-submit the high-resolution versions of Supplementary Figures S1 and S2 to ensure optimal readability. Additionally, we have added a new table (Table 1) in the revised manuscript to present the key data from these figures.
Section 3.4. When describing statistically significant differences in microbial communities, the authors attribute this interpretation to PCoA results. This constitutes a methodological inaccuracy, as PCoA serves solely as a dimensionality reduction and visualization technique without inherent statistical testing capabilities. The text should be revised to reference PERMANOVA for significance testing.
Response: Thank you for this crucial observation. In the revised manuscript, Section 3.4. has been corrected to explicitly reference PERMANOVA as the statistical method used to test for significant differences in microbial communities, please see Lines 288-290, Lines 297-298, Lines 311.
Figures 1, 3, 4, and 5. The authors should increase the font size.
Response: Thank you for your suggestion. We have re-submitted Figures 1, 3, 4, and 5 with enlarged font sizes to enhance readability.
Section 3.6. In presenting multiple correlation analyses, the authors have not indicated whether statistical adjustments were implemented to control for Type I error inflation inherent in simultaneous hypothesis testing. It is methodologically imperative to specify which multiplicity correction technique (e.g., Bonferroni, Benjamini-Hochberg false discovery rate) was applied to the correlation matrices.
Response: Thank you for this important methodological observation. In the revised manuscript, we have clarified that to control for Type I error inflation during simultaneous hypothesis testing, the Benjamini-Hochberg false discovery rate (FDR) correction was applied to the correlation matrices. This approach is widely recognized as appropriate for genomic and microbial community analyses, where multiple comparisons are common, as it balances the need to reduce false positives while maintaining statistical power.
The relevant description has been added in Lines 365-367 of Section 3.6. We appreciate your guidance in strengthening the rigor of our statistical methods.
Discussion.
L 481. The authors should report the specific mechanisms or microbial taxa responsible for enhanced organic matter degradation in LFC systems.
Response: Thank you for this insightful suggestion. In the revised manuscript, we have supplemented detailed information in Lines 461-463 as follows: LEfSe revealed enrichment of organic matter-degrading microorganisms in LFC sediments, with fungal (Trichoderma, Aspergillus) and archaeal (Methanosarcina) taxa as key contributors.
We thank you for pushing us to deepen this critical aspect of our discussion.
L 506. The authors should specify which nutrient thresholds (TN, TP) are considered critical for driving viral diversity in freshwater systems.
Response: We agree with you completely that establishing precise nutrient thresholds is important for understanding the mechanistic relationship between nutrient concentration, microbial productivity and viral diversity. Currently, there is no consistent academic consensus on the critical TN and TP thresholds that would influence the viral diversity in freshwater systems. While our results showed a wide range of nutrient concentrations and demonstrated positive correlations between nutrient levels and viral diversity via correlation analyses, gradient experiments were not performed. Therefore, universally applicable TN/TP thresholds cannot be directly extrapolated from our data at this time.
L 528. The authors should discuss the practical implications of suppressed Herpesvirus activity in LFC fish populations, especially concerning aquaculture productivity and disease management.
Response: Thank you for this valuable suggestion. In the revised manuscript, we have supplemented relevant discussions (Lines 511-515) as follows: The inhibition of herpesvirus mediated by LFC does not merely augment production by diminishing mortality rates; it also transforms disease management paradigms. This is achieved by decreasing reliance on chemical controls and enriching antagonistic microbiota, which competitively inhibit viral proliferation.
We thank you for prompting this discussion, which strengthens the practical relevance of our findings.
L 546. The authors should elaborate on the specific mechanisms by which lotus roots influenced sediment properties.
Response: Thank you for this insightful suggestion. We appreciate the opportunity to elaborate on the specific mechanisms by which lotus roots influence sediment properties in the LFC system. In the revised manuscript, we have supplemented detailed explanations in Lines 531-535: Lotus roots act as pivotal ecosystem engineers within sediment systems, exerting their in-fluence through three interconnected mechanisms: (1) physical restructuring that en-hances sediment porosity and promotes particle aggregation; (2) chemical regulation that modulates redox potential and mediates nutrient fluxes; and (3) biological facilitation of microbial consortia, which drive organic matter degradation and pollutant transformation [86-88].
The authors should discuss how their findings can be scaled or implemented in larger, real-world aquaculture operations.
Response: Thank you for this valuable suggestion, which highlights the practical significance of our research. In the revised manuscript, we have added a discussion on scaling and implementing our findings in large-scale aquaculture operations, please see Lines 650-667.

Reviewer 4 Report
Comments and Suggestions for Authors
This MS is devoted to the study of microbial communities in two ecosystems - rice and lotus communities. Rice and lotus agricultural farms have developed very widely throughout the world over the past 10 years, especially in the countries of the Asia-Pacific region. The study conducted by the authors is a comprehensive study with an emphasis on metagenomic studies.
I must note that the authors adequately selected the methods and carried out a high-quality metagenomic analysis. A detailed description of the method allows the reader to double-check the data obtained in the current MS.
In the Results chapter, the authors provided irrefutable evidence of the diversity of the microflora of the two ecosystems. The a and b analysis proves that fisheries using lotus ponds are much more effective than for rice ponds.
In the Discussion chapter, the authors very competently discussed their data with the literature. All the conclusions made by the authors are logical.
I have to note that I was personally pleased with the structure of the manuscript and the ease of reading it. Unfortunately, I could not find any minor comments on the text😊 The text is well proofread and has a high level of English. I believe that this MS can be published in the journal Biology in its current form.
Author Response
Comments and Suggestions for Authors
This MS is devoted to the study of microbial communities in two ecosystems - rice and lotus communities. Rice and lotus agricultural farms have developed very widely throughout the world over the past 10 years, especially in the countries of the Asia-Pacific region. The study conducted by the authors is a comprehensive study with an emphasis on metagenomic studies.
I must note that the authors adequately selected the methods and carried out a high-quality metagenomic analysis. A detailed description of the method allows the reader to double-check the data obtained in the current MS.
In the Results chapter, the authors provided irrefutable evidence of the diversity of the microflora of the two ecosystems. The a and b analysis proves that fisheries using lotus ponds are much more effective than for rice ponds.
In the Discussion chapter, the authors very competently discussed their data with the literature. All the conclusions made by the authors are logical.
I have to note that I was personally pleased with the structure of the manuscript and the ease of reading it. Unfortunately, I could not find any minor comments on the text. The text is well proofread and has a high level of English. I believe that this MS can be published in the journal Biology in its current form.
Response: Thank you very much for your thorough review of our manuscript and your extremely positive and encouraging comments. We are truly grateful for the time and effort you have dedicated to evaluating our work on microbial communities in lotus-fish ecosystems.
We are particularly pleased to hear that you approve of our selection of methods and the high-quality metagenomic analysis, as well as the detailed description of the methods which enables readers to verify our data. It is also very rewarding to know that you found the evidence of microflora diversity in the Results section compelling and that our conclusions in the Discussion are logical and well-supported by the literature.
Once again, thank you for your valuable input and enthusiastic support.
